# Intergenerational Esthetic Co-Creation Program for Promotion of Community Creative Aging

**DOI:** 10.3390/healthcare11040516

**Published:** 2023-02-09

**Authors:** Hsiu-Ching (Laura) Hsieh, Chun-Wei Liu

**Affiliations:** Department of Creative Design, National Yunlin University of Science and Technology, Yunlin 64002, Taiwan

**Keywords:** intergenerational co-creation, esthetic workshops, creative aging

## Abstract

The world is facing rapid global aging. Global countries are concerned about the development of aging societies and related topics ranging from successful, healthy, and active aging in the past to the current creative aging (CA) perspective. However, in-depth research on applying esthetics to promote community CA in Taiwan is lacking. To address this deficiency, the Hushan community in Douliu City, Yunlin County was selected as the research area, and the CA perspective was adopted to promote community CA through multi-stage intergenerational esthetic co-creation (IEC) workshops. A model for applying IEC workshops to promote CA was constructed. Using the action research approach, community CA enabled the elderly to identify with their inherent values, opening new possibilities for the provision of elderly social care. This study explored the psychological impacts of implementing IEC workshops on the elderly, analyzed their interactions with peers and youth, helped the elderly review their lives, analyzed relevant data to construct a practical model for applying IEC workshops to promote CA, and provided the data collected during the multi-stage applications of CA and an IEC model for promoting CA as a reference for future researchers, thereby opening new possibilities for sustainable care in aging societies.

## 1. Introduction

### 1.1. Research Background and Motivation 

Aging is a silver wave that cannot be ignored in today’s society. The World Health Organization defines aging as a privilege, an achievement, and a challenge and has warned that aging is an unprecedented challenge and the biggest task that individuals and societies must be prepared to face [1]. Other than the gradual weakening of the physical body, the elderly must also adapt to issues such as psychological loneliness and changes in social roles and status. Hence, it is important to find ways to help them adapt to living in old age, maintain their quality of life, and ensure that their lives are fulfilling and meaningful. The United Nations Second World Assembly on Aging in 2002 highlighted that the basis of a humanized aging process is to provide the elderly with opportunities to participate in creative cultural and artistic activities.

Presently, various countries are paying attention to the concept and development of creative aging (CA) [2]. With the growing elderly population, more countries have begun paying attention to the developmental aspects of aging. Both theoretical and empirical studies have concluded that art plays an important role in the lives of the elderly and provides physiological benefits. Specifically, the numerous physiological, psychological, and social benefits improve their quality of and subjective satisfaction with life. The elderly are subjects capable of development and creativity from the perspectives of successful, active, and creative aging. Encouraging the elderly to participate in art activities, unleash their creativity, and activate their brain cells has become a global trend.

Yang [2] pointed out that the existing research on art and quality of life among the elderly in Taiwan has mostly regarded the elderly as persons in need of healing and has failed to treat elderly people as developmental and creative beings in line with the concept of active aging [2]. Hence, adopting the CA approach and emphasizing the use of art as a medium to provide the elderly with challenging and engaging activities has been recommended [3]. According to Lin and Lee [4], community empowerment is an important reference indicator for a country’s socioeconomic development. The cultivation and utilization of community bases is actually an important topic for aging in place. A review of the previous literature and the current situation revealed a lack of studies in Taiwan from the CA perspective as well as a dearth of community-focused esthetic solutions. The present study sought to help bridge those research gaps. This study, therefore, focused on the community with the aim of guiding the elderly to become part of a community and to interact in order to enhance their social participation.

The CA perspective emphasizes long-term, diversified, and structured esthetic co-creation workshops that allow the elderly to enhance their self-identity, promote interpersonal interaction in the later stages of life, and further improve people’s quality of life in their later years. In particular, CA focuses on the use of intergenerational esthetic co-creation (IEC) as a tool for community participation among the elderly. Participation in life esthetics can promote the elderly’s familial and community lives, enrich life in old age, and improve physical and mental health. In turn, elderly people’s quality of life is enhanced, and elder–youth interaction increases. 

### 1.2. Research Objectives 

To explore the impacts of implementing IEC workshops on the psychology of the elderly.To analyze the impacts of IEC workshops on promoting interactions and exchanges between the elderly and their peers as well as with younger generationsTo analyze the use of IEC workshops to help the elderly reflect on their life journeyTo construct a model for applying IEC workshops in order to promote CA.

### 1.3. Research Scope 

The Hushan community in Taiwan’s Yunlin County was selected as the research area. This community, located in the eastern suburb of Douliu City, has approximately 800 residents who live honestly and simply. It is an agricultural settlement in the mountains with rich land resources, century-old trees, old houses, and beautiful scenery. The residents’ main livelihood when they were young and able-bodied comprised of agricultural activities. Having been engaged in the agricultural specialties of sour bamboo shoots and dried longan fruits, they have formed a living circle with strong humanistic sensibilities.

This community was selected for several reasons. First, it is facing the problem of aging and the outflow of young people, meaning that most elderly live alone. Second, the community development association has paid attention to and actively dealt with the aging problem in recent years through the establishment of intergenerational dining halls and regular social worker visitations. This has strengthened community cohesion regarding the management of the aging problem. Finally, despite being a small hinterland, the community possesses numerous landscapes with historical and cultural characteristics.

## 2. Literature Review

### 2.1. From Successful Aging to Creative Aging: A Global Developmental Trend

Psychologists Baltes and Baltes [5] have proposed that, for the individual, the key points for achieving successful aging are choosing and adapting to a suitable life pattern and strengthening one’s potential through educational, motivational, and health-related activities, as doing so enables one to have a functional and productive life after making the requisite adjustments, and to work toward goal alignment without losing oneself. Healthy aging goes beyond the health of the elderly and not only considers the actual impacts of disease on individuals’ functioning and well-being but also provides ways to measure and promote it [6]. Therefore, by definition, healthy aging refers to aging in such a way as to develop and maintain the functions required for healthy living in old age. This involves two critical concepts: inner health and functional optimization. 

The European Union [7] has defined “active aging” as the elderly continuing to participate in the formal labor force and unpaid productive activities (e.g., caring for family members and/or volunteering), thereby leading healthy, independent, and safe lives. The active aging index (AAI) has also been proposed and incorporates four aspects: employment, social participation, healthy independence, and safety. With supportive policies, the elderly can enjoy life without facing age discrimination. This will allow them to better adapt to the social environment, be assured of their physical, physiological, and mental health, and be economically independent. Additionally, goals related to the appreciation of assets and security can be achieved through an appropriate government financial management system [8].

The Creative Aging concept was first proposed by renowned American psychiatrist and expert gerontologist Dr. Cohen [9]. It suggests that the elderly be allowed to continue developing their entire brain through balanced static–dynamic social activities, social participation, and art activities. It also advocates that the elderly be encouraged to awaken their inner artistic potential in the latter half of life by attending “multifaceted and structured art courses” that will help them to establish a strong self-identity, reach a high level of life satisfaction, and enjoy fulfilling relationships. 

Among the elderly, creativity basically develops in the latter half of life according to one of the following three models [9]: The first is the activation of the creativity model, which purports that the elderly are able to freely develop their interest in art and summon the courage to experiment and try new things after finally being relieved of societal pressures and responsibilities. Under the conditions of internal and external freedom, they can tap their years of experience to activate the depths of their creativity. The second is the creativity continuation or change model, which acknowledges that some people find a channel for their creativity during their youth, and in their senior years, after experiencing the liberation, summary, and encore life stages, their mentality changes, stimulating new modes of creative expression. Third is the creativity inspired by the loss model, which purports that the inevitable loss and negative emotions aging brings are transformed into a medium that stimulates creativity through creative channels. According to this model, seniors’ undervalued potential is brought out by a sense of crisis and loss, ushering them into the all-cause creative stage. Previous studies have indicated that creativity can blossom and bear fruit at any age and all stages, with the creative achievements of older individuals being even more profound and abundant.

The CA concept was adopted to guide the design of the IEC workshops in this study. The well-structured and comprehensive workshop content was developed to guide the elderly to explore their creativity and potential in the latter half of their life. CA has three positive effects on the elderly. The first is a sense of control. Through engagement in activities, the elderly gain a sense of proficiency and control, which makes them feel capable and stimulates more confidence and energy to attempt new endeavors. These positive feelings in turn improve their immune system, which contributes to physical health. The second is social participation. Having active social and interpersonal relationships in the latter half of life reduces stress and keeps the immune system functioning well. Membership in such a network also means that the elderly have greater support available in times of loss. The third positive effect emanates from the fascinating nature of art. Sustained perseverance is inherent to the act of creation, and the process is pleasant, interesting, easily obtainable, and achievable. Hence, art draws people to invest their time, and activities sustained over the long term are more valuable and effective than one-off experiences [9].

### 2.2. Related Theories and Research on Applying Art and Esthetics Workshops to CA

Liu [10] suggested that art reveals the true nature of the world, offers greater enlightenment about life, and brings happiness and well-being by virtue of its comforting nature. In other words, people observe the beauty of art and experience greater happiness through the process of expressing ideas, phenomena, and memories about beauty and kindness as life’s essence. This is similar to Cohen’s idea regarding the attractiveness of art and esthetics, which gives people a sustained, pleasant, and interesting feeling.

In line with the above, actual cases related to the application of art and esthetics to aging and the impacts on the elderly are discussed in this section. One of the best examples is Meet Me at Albany, a case study cited in the UK government’s 2016 white paper [11] on culture. The Meet Me at Albany program [11] aimed to (i) build a community comprising elderly who are healthy, disabled, or living alone and establish new connections among them through joint participation in art activities; (ii) formulate plans to develop artistic skills in that community of the elderly; (iii) identify and support emerging mature artists, and bring new purpose and meaning to the lives of lonely elders through their participation in art practices, creating situations that allow for recognition of their life experiences; and (iv) provide opportunities for younger elders to find new ways of supporting older and lonely elders in the context of community activities through participation in art practices.

Havighurst [12] proposed activity theory, which posits that the elderly seek to maintain the lifestyle they had when they were middle-aged. Regarding the aging process, activity theory suggests that continued participation in physical, psychological, and social activities will give aging persons a sense of satisfaction [13]. With the promotion of the new CA trend, people are gradually valuing the benefits of art programs for the elderly living in institutions and communities.

The impacts of art and esthetics programs on the elderly can be explained by the following three aspects: The first is physiological health. Benson [14] argued that the creative process stimulates special areas of the brain to release endorphins and other neurotransmitters. In other words, art activities promote the development of cerebral nerve cells, boost cerebral cells and the immune system, and switch the body to a more relaxed state [15]. Regarding the second aspect, mental health, Fan [16] found that the elderly who participate in creative activities were more active, gained satisfaction from achieving goals, had a sense of accomplishment, and felt that there were things worthy of pursuit [17], which allowed them to achieve high levels of self-actualization per Maslow’s theory. The third aspect is social participation. People have higher levels of interaction with their surrounding environment during the creative process, and the elderly who participate in artistic activities have a better mental state and more connections with others [18]. Moreover, the diversity and symbolism of art provide the elderly with abundant opportunities to convey information and integrate their thoughts and emotions, which effectively helps them face uncertainties in life [2].

### 2.3. Cases of the Application of Art and Esthetics Workshops to Promoting CA

CA indicates that art and esthetics programs enable the elderly to achieve positive physical and spiritual development. This is because the particularity of art and esthetics lies in its metaphorical and symbolic functions, and the creative process demonstrates creativity and conveys life experiences and stories [19]. Hence, art and esthetic programs for the elderly have been developed and implemented in various countries. Three such programs will be analyzed and discussed next.

#### 2.3.1. Legacy Artwork

This is a service proposal planned and developed by Purlstein [20], founder of the New York-based non-profit organization Elders Share the Arts (ESTA). It combines oral history and the CA concept in response to the issue of the elderly lacking social interaction and participation while also reinforcing the image of the elderly as creative and energetic. The model involves theme-based group activities and presents the life stories of the elderly in concrete ways through artistic creations, such as paintings and handicrafts. The purposes are to respect elders’ self-expression, let them experience new things to compensate for their regrets, and fully utilize visual art to create good times and happy memories. Sharing their life history gives elders the opportunity to reinterpret their lives through feedback from others and forge closer relationships.

#### 2.3.2. Intergenerational Art Program

This educational program facilitates the meaningful and continuous exchange of resources and learning channels between youth and the elderly for personal and social benefit. It allows elders and youths to learn together in various contexts while encouraging, caring for, interacting, and cooperating with one another and also facilitates the meaningful exchange of social experiences [21]. Intergenerational interactions comprise the critical element of this program, with elders and youths connecting through co-creation, thereby improving the elderly’s quality of life.

#### 2.3.3. A Lifetime of Looking

This program [22] assists the elderly with self-integration through their active recall of past achievements and failures. Reviewing one’s life serves several functions for the elderly, namely, imparting, self-affirmation, integration, and preparation for death. Psychologists believe that the process allows the elderly to reminisce about the past [23,24]. People reminisce more frequently as they grow old; for the elderly, the act of reminiscing clarifies their thinking and helps them regain their self-confidence and face life with a positive attitude. In Taiwan, the Hondao Senior Citizen’s Welfare Foundation [25] launched a life review program for community elders and found that participants were able to reorganize their life history thematically, which led them to discover their dreams. Methods used include educational courses, drama, sports, and other community care programs that have helped the elderly achieve self-realization. This in turn stimulated their vitality and endowed them with autonomy and dignity.

### 2.4. Summary of the Literature

The literature cited above indicates that although CA is a global trend, relevant practical experience and research on the application of esthetics to the promotion of community CA in Taiwan are lacking. This study applied Cohen’s CA theory [9] to the design of an art program for implementation in the Hushan community, the culture and life context of which were adopted as the core values of the esthetic workshops. The life context of the community elders and the concepts of Legacy Artwork, Intergenerational Art Programs, and A Lifetime of Looking were integrated into the planning of the workshop content. Additionally, elders and their companions would be invited to participate in the workshop design. In sum, the IEC workshops were used to construct a sustainable CA development model that emphasizes the vitality of life.

## 3. Methodology 

The action research method—in the form of IEC workshops—was adopted to stimulate elders’ vitality and creativity. Guided by the workshop content, participating elderly could interact in their community environment with peers and youth and be drawn into social participation. The action research method emphasizes periodic critical reflection, adjustments based on the problems faced, and concomitant improvement processes. Participants formulated appropriate solutions to the problems they encountered during the workshops. Evaluation, discussion, feedback, and remodification were also performed and served as the basis for improving the next action cycle. Hence, the program was implemented in a real situation, which was the basis for review and analysis, followed by the formulation of new countermeasures. Problems were verified during the continuous cycle of linking past experiences and future actions, eventually leading to solutions and ultimately improving the theory’s feasibility [26].

### 3.1. Research Participants

The participants were elderly (aged 65 years and above) living in the Hushan community.

### 3.2. Research Process

Details of the research process are elaborated below.

#### 3.2.1. Define the Problem 

It was ascertained from research observations and the literature review that the autonomy and dignity the elderly need are often ignored during the provision of elderly care in Taiwan. The numerous existing service agencies and communities in this sector provide basic care, curriculum arrangement, and companionship for the elderly; however, elderly people’s creativity, social connections, and life journey are often neglected. Additionally, there is a lack of community art and esthetics programs designed with the elderly in mind.

#### 3.2.2. Formulate the Action Plan 

This task involved four main implementation stages, namely, three art creation stages, with the fourth stage consisting of an esthetics exhibition, as shown in Table 1. Step 1 was to design the framework of the IEC workshops. The research team visited and conducted field surveys in various communities in the early stages of the study to gain an in-depth understanding of their characteristic landscape and cultural customs, as well as the elderly residents’ backgrounds and personalities. The observed situation and esthetic and creative thinking were combined to plan a suitable framework for workshops that would cater to the community members. Step 2 involved securing the necessary support to implement the IEC workshops. Step 3 entailed evaluation of the implementation venue and related planning, with the Hushan community eventually being selected.

Step 4 entailed seeking community members’ assistance. The research team cooperated with the Hushan Community Development Association to conduct in-depth research and implement the workshops. Community members and social workers assisted us by contacting elders for scheduling, arranging the workshop venue, and handling the necessary equipment. During the course, they also assisted the participating elders and monitored their physical and mental states. Overall, they played the role of workshop facilitators.

#### 3.2.3. Reflect before Re-Planning

This study was conducted according to the action research method, involving the four spiraling and cyclical steps of planning, action, observation, and reflection (Tsai, 2000). The contents and process were reviewed at each implementation stage, and continuous revisions were performed. There were also reflections and re-planning of the implementation process throughout the multi-stage workshops, which served as the basis for improving the content of the workshops to be held at the next stage.

#### 3.2.4. IEC Program Implementation 

The community’s cultural landscape, the elders’ needs, and practical problems were identified through field investigations conducted by the research team. The findings informed the workshop content to ensure its appropriateness for the community elders.

#### 3.2.5. Compile Information on IEC Program Implementation to Promote CA

During the research process, data were collected via semi-structured interviews (see Table 2). Records of the problems identified at each stage, the corresponding reflections (Table 3), and on-site workshop records (Table 4) were also compiled. Some of the IEC artistic creations are shown in Figure 1, Figure 2 and Figure 3. The records and feedback were coded according to their attributes and analyzed to arrive at conclusions and recommendations. 

The semi-structured interview objectives for this study contained 15 senior participants, 2 workshop facilitators, and 3 workshop staff. The interview of senior participant was arranged from late December 2021 to end of January 2022 (the period after the exhibition at the fourth stage of intergenerational esthetic co-creation workshop). The independent and private lounge in Hushan Community Activity Center was selected for the interview in December. Each interviewee signed the informed consent form before the interview. One person was interviewed at a time for about 1.5 h. The interview in January 2022, was conducted in 9 senior participants’ homes. Nine senior participants expressed that they could more comfortably speak out their feelings at home. The interview questions were based on the questionnaire designed for this study. The questions were designed based on Havighurst’s Activity Theory [12] to understand the psychological, physiological, and social effects of the esthetic workshop on senior participants through the perspectives and ideas after participating in the workshop. The interview questions are designed in Section A.1. 

Two facilitators were interviewed in late December 2021 in the independent and private lounge in Hushan Community Activity Center. One person at a time was interviewed for about 2 h. The interview questions were based on the questions designed for this study. The interview questions covered three major dimensions of practical experience, concept identification, and benefit outlook to apply artistic esthetic workshop to implement creative aging. The interview questions are designed as Section A.2. 

Three staff were interviewed as the improvement reference for the planning and execution of next esthetic workshops. The interview was done in the design laboratory (Room DA514) at National Yunlin University of Science and Technology for about 1.5 h. The interview content and points contained the appropriateness of content planning to implement creative aging through esthetic workshop, challenge of action practice, and reflection to achieve creative aging. The interview questions are designed as Section A.3.

#### 3.2.6. Knowledge Sharing and Holding an Exhibition 

At the final stage of the workshops, a public exhibition was held at the Hushan Community Activity Center, and the works the elders created throughout the workshops, as well as their various sentimental objects, were displayed. Other community members, including the participating elders’ relatives, were invited to attend the exhibition, and the elders were encouraged to share their creations and past memories and provide feedback on the workshops.

## 4. Results

The developed framework indicating the various stages of the IEC workshops is shown in Table 1, and the data coding of the interview results is given in Table 2. Records of the problems identified at each stage and the post-reflection responses are shown in Table 3, and on-site workshop records corresponding to the various stages are shown in Table 4. Some of the works the elders and youths co-created during the workshops are shown in Figure 1, Figure 2 and Figure 3. The data collected during the study were cross-analyzed and are discussed in the next Section.

## 5. Discussion 

The content of the interviews with the elders who participated in the workshops, as well as that of the interviews conducted with the facilitators (volunteers) and staff (graduate students who volunteered as companions for the elders), are summarized and discussed in this section.

### 5.1. Analysis of Interviews with Elderly Workshop Participants

#### 5.1.1. Numerous Positive Effects

The elders derived happiness and a sense of accomplishment from participating in the IEC workshops. They enjoyed the creative process and new experiences. Participants remarked as follows: “I was happy when creating the artworks” (E1-I-Q1-1021), “I felt formidable… and I liked what I created very much!” (E3-I-Q1-1021), and “I felt very happy and good when looking at my creations. I was interested in the handicraft work and [in] drawing” (E12-I-Q1-1021).

Additionally, the elders said that participating in the workshops increased their degree of interaction with their peers: “I chit-chat with everyone when I am here; [we talk] about our current situation and life in general” (E13-I-Q2-1122).

As evidenced by the following comments, the elders derived satisfaction and joy from recalling their life history and organizing and sharing their memories during the workshops: “Looking at a photograph when I was an 18-year-old young lady… but I am already a grandmother now” (E5-I-Q6-1222); “Sharing stories is a very good idea. [It used to be that] only I knew how difficult my life was. Now, more people know about the past” (E2-I-Q6-1223).

#### 5.1.2. Psychological and Physical Relief

Workshop participation was a means for the community elderly to alleviate their psychological pressures and temporarily forget about their physical ailments, as evidenced by the following comment: “I am foolishly happy when [I’m] chit-chatting with friends. [While I’m] laughing away, I do not think about my pain and illness” (E11-Q7-1226).

#### 5.1.3. Social Impacts of Elders’ Participation

During the interviews, the elders said that they looked forward to interacting with the youths and that family support motivated their engagement in community activities. Participating in the workshops and interacting with others helped them block negative thoughts and refocus on life, as evidenced by the following comments: “Although my children do not live with me, they are supportive when they know that I am attending the workshops” (E 8-Q8-1227); “My family is quite supportive. I do not think that what I have made looks good, but my daughter-in-law said that I am skillful!” (E12-Q8-1227); “I have become happier and more outgoing since attending the fun courses” (E2-Q8-1227).

#### 5.1.4. Able-Bodied Elders’ Contributions 

Some elders expressed their willingness to contribute to the community. Some who live alone and are sufficiently able-bodied to participate in community activities remarked as follows: “I feel that I am not that old, so I come here to offer my assistance to the other elderly people” (E9-Q9-1228); “It is fun to come here to chit-chat since I am free. Plus, they teach us how to play various games” (E2-Q9-1228); “I grow vegetables in my backyard. I eat some myself and share the rest with others” (E12-Q9-1229).

### 5.2. Analysis of Interviews with Volunteer Workshop Facilitators

#### 5.2.1. Need for Careful Observation 

The elderly generally showed a passive attitude toward community activities and instinctively resisted unfamiliar things. It is therefore important to closely observe their behavior, including their reactions to prompts/invitations. On this matter, the volunteers remarked as follows: “We need to be patient when inviting the elders to participate and must keep trying” (F1-I-Q1-1221); “When the elders are not interested or have no confidence in a task, their body will feel the pressure, and they become worried that they will not do well. When that happens, they will not want to try again” (F2-I-Q3-1224).

#### 5.2.2. Slowed Aging Rate

The volunteers found that the workshops motivated the elders to venture out of their homes, which slowed the rate at which they aged. One volunteer commented as follows:

“There were noticeable changes in the elders after the workshops. When some of them know the teacher is coming today, they show up on time on their own accord, without needing to be repeatedly persuaded like in the past. The workshops help them go outside their homes and learn new knowledge, which slows the rate of aging” (F1-I-Q5-1221).

### 5.3. Analysis of Interviews with Workshop Staff (Companions)

#### 5.3.1. Adopting the Elders’ Perspective

As reflected in the following comments, workshop content and implementation modes should reflect the perspective of the target elderly participants. Art media should also be chosen based on its suitability for elderly participants. “We completed the more difficult part of the production process first, which made it easier and more enjoyable for the elders when the serigraph was finally printed” (S1-Week16-01). “The elders in the community were highly receptive to the courses. They were also very familiar with one another and enjoyed chit-chatting. Time slots [were] specially [designated] for sharing, [and] the elders were encouraged to discuss topics among themselves” (S3-Week16-01).

#### 5.3.2. Elders’ Acceptance of Art

During the IEC implementation process, it was noted that the workshops helped the elders accept art. Positive elder–youth interaction was also noted, with the former gaining a sense of accomplishment and opportunities for emotional catharsis. However, as noted in the following comments, several difficulties were encountered during the process. “Obstructions to the co-creation process [arose] when the elders were unfamiliar with the companions” (S2-Week18-02). “More time must be spent communicating and getting familiar with the elders in the early stage. Only then can their mentality shift from passivity to active learning”. (S2-Week18-03).

#### 5.3.3. Display of Creativity

As evidenced by the following quotes from the staff, the elders developed a more proactive approach during the workshops, which effectively activated their vitality and creativity. “Initially, the elders were relatively introverted and passive. They only made attempts when we guided them. Later on, after they became closer to us, we found that they would take the initiative to do things on their own” (S2-Week18-04). “The workshops activated the elders’ hands and brains. They were also able to share their life experiences during the creation process, which stimulated [their recollection of] their life [including] memories from the past” (S3-Week18-03).

#### 5.3.4. Need for Sustained Implementation

As evidenced by the following remarks from the staff, the workshops must be sustained over a long time for them to be effective. During the process, it is also necessary to praise the elders regularly and adjust the difficulty of the tasks in a timely manner. “The workshops must be sustained for a long time to be effective. Understanding the community and the elders’ lives at the early stage was critical to ensure that the workshops better met their needs” (S2-Week20-02). “The elders were motivated during the creation process when they were praised at the appropriate time” (S3-Week20-02). “When the tasks were too difficult and the elders repeatedly failed to complete them, we quickly switched to other tasks. This required careful observation and empathy on our part” (S2-Week20-03).

## 6. Conclusions

Data from the multiple IEC workshop cycles held in the Hushan community, the content of the interviews with the various workshop actors mentioned in Section 5, and the characteristics of the elders’ creations were compiled and used as the basis to draw the following conclusions in direct response to the study’s objectives.

### 6.1. The IEC Workshops Enhanced the Elders’ Self-Confidence and Facilitated Interaction

#### 6.1.1. Satisfaction and a Sense of Achievement Derived from Creating Enhanced the Elders’ Self-Confidence

Cross-analysis and inductive analysis of the implementation of the multi-stage workshops and the interview data indicated that because creating esthetic art evokes fascination, this characteristic allowed the elders to temporarily dispel their negative emotions, encounter novel experiences during the creation process, stimulate their creativity and potential, and gain psychological satisfaction. They exchanged opinions and experiences with their peers and the youths while creating during the workshops. These interactions allowed them to rebuild their confidence and reach a state of happiness while also increasing their willingness to continue creating art.

#### 6.1.2. Holding the Esthetic Exhibition Gave the Elders a Sense of Accomplishment

Holding an exhibition was a component of the workshops. Participating in an art exhibition allowed the elders to discover their own potential and value and benefit from the sense of being well-grounded that comes from having a clear identity. They orally shared their memories and creative ideas. In the process, they gained a sense of accomplishment and demonstrated self-confidence and self-affirmation.

#### 6.1.3. The Workshops Provided Opportunities for Social Participation and Expanded the Elders’ Interpersonal Relationships

The interview content revealed that the workshops encouraged the elders to venture out of their homes and make contact with the outside world. This helped them break through their psychological and physical limitations, participate in social activities, and expand their interpersonal network through relationships with their peers as well as with youths. The creative process provided many opportunities for the elders to share their experiences and get to know their interlocutors. Participating elders also listened to other people’s stories and gave feedback. Such sharing allowed others to better understand the elders’ needs. Mutual empathy and caring also contributed to the establishment of closer social relationships.

#### 6.1.4. The Workshops Changed the Elders’ Mentality from Passive to Active

After the elders were familiarized with the format and flow of the workshops and formed emotional connections with the youths, workshop participation gradually became an important event for the elders and one of the focuses in each participant’s life.

### 6.2. The IEC Workshops Promoted Interactions between the Elders and Their Peers and the Youths

#### 6.2.1. Mutual Learning through Elder–Youth Interaction

During the workshops and various activities, the researchers observed the elders’ interactions from the sidelines and found that the elders and the youths regarded one another as partners. They discussed solutions to the problems that arose, shared their experiences, and engaged in emotional exchanges while completing the artworks. The elders’ suggestions were incorporated into the workshops, and the youths applied the elders’ wisdom to practical matters. At the end, the elders and the youths worked together to hold an art exhibition and celebrated the fruit of their combined efforts. They also formed close partnerships while learning from one another.

This situation corresponds to Roberto’s (2015) model of theoretical intergenerational experiences, which covers the in-common experience, parallel experience, contributory occasions, and interactive sharing. The instances of intergenerational co-learning during the workshops can be equated to interactive sharing. The most obvious learning mode was through interpersonal exchange, where the content included experiences, ideas, and emotions, as well as being able to listen and respond to others.

#### 6.2.2. The Youths Energized the Community and, through Innovation, Helped the Elders Develop Their Potential 

During the interviews, the elders reported thoroughly enjoying interacting with the youths and noted that they felt that they had much to share with them. When the youths helped the elders learn how to use innovative media, the latter came into contact with new things, which enhanced their mental flexibility and unlocked their potential.

### 6.3. The IEC Workshops Promoted Interactions between the Elders and Their Peers and the Youths

#### Nostalgic Prompts Guided the Elders to Review and Synthesize Their Lives

Analysis of the various stages of workshop implementation and interview data revealed that the elders recalled their youthful appearances and happy moments when viewing and working with old photographs and objects. Without the guidance of the workshop context, they may not have had the opportunity to revisit those photographs and objects, meaning that those memories might have otherwise been permanently sealed off and easily forgotten over time. Therefore, during the workshops, nostalgic prompts (in the form of sentimental objects) helped the elders re-examine their memories and synthesize their lives. Through reviewing and sharing their life trajectory during the workshops, the participating elders found resonance in one another’s experiences given their immersion in a peer group from the same generation with common life experiences. Notably, mutual support was one of the important sources of happiness (Hwang and Yang, 2006). Additionally, the content that the elders shared allowed the youths and other community members to better understand the elders’ life history and develop empathy for their behaviors and perspective.

### 6.4. The IEC Program Was Successfully Developed and Applied to Promote the CA Model

The IEC program was successfully integrated into the community in this study, with the IEC workshops serving as life development tools and an impetus for citizen participation, ultimately promoting elders’ community life. The program also facilitated connections and communication between the elderly and youths. Several considerations are important for the implementation of such a model. First, community resources and the target elders’ lifestyle/background should be thoroughly analyzed before the development of the esthetic workshops, which should take community life as their core component (i.e., community-based collaboration and multi-role participation). The experiential feedback received from the elderly participants and the facilitators/companions was collected over the course of the multi-stage program implementation, and adjustments were made periodically throughout the various program cycles. This facilitated effective CA interactions among the youths and the elderly participants who enjoyed engaging in, discussing, responding to, and experiencing esthetic co-creation, as well as participating in the process and understanding the connotations of life culture and far-reaching underlying meaning. The model for promoting CA through the IEC workshops is shown in Figure 4.

Community collaboration/cooperation was a necessary condition for the implementation of CA. The considerations for identifying a suitable community include whether the community and the implementation venue possess sufficient human and material resources and whether the community elders have the willingness, energy, and time to participate in the proposed activities. A mutual cooperative relationship is essential to maximally effective program implementation.

Before implementing the IEC program, given that the community’s life context constitutes the program’s core, the staff must, through investigations and interviews, gain a working understanding of the community elders’ life context and ensure that program planning aligns with their preferences and habits, which should be integrated into the content of the esthetic workshops. During the process, it is also necessary to continuously review and modify the purpose of the action and the method (assessed based on the results) while adopting the participating elders’ perspective to ascertain their degree of acceptance of the workshops and the degree to which they like and enjoy the content to determine whether there is a need to terminate or modify specific program items. The action strategies should be modified on a rolling basis according to staff reflections, and problems that arise at the various stages should be recorded so that the records may serve as the basis for modifications, with the ultimate goal of motivating the elders to participate and interact.

Multi-role participation is required for the proper implementation of the IEC workshops in order to achieve CA. The various roles are explained below.
Community Elders: An esthetic program suitable for elders should be planned after holding related discussions with them in order to cater to their specific needs. The creation and sharing processes during the workshops can be used to help participating elders review their life history and expand their interpersonal network. Their participation also provides momentum for the youths and the community so that the former can co-learn through the elders’ life experiences, and community cohesion can be strengthened through the elders.Volunteer Facilitators from the Community: Facilitators play an indispensable role in communication. They lead the initial investigations, meetings, and discussions regarding the workshop contents and provide the history of the community and its folk customs, which are elements for planning the workshops. During program implementation, they serve as a bridge between the elders and the staff and researchers. Given their familiarity with the elders, facilitators can encourage them to continue participating in the workshops. They also sometimes provide assistance during the courses and monitor the elders’ physical and psychological conditions.Staff/Companions for the Elderly: These are youths who act as companions for the elders during the workshops. Their duties include conveying esthetic skills to the participating elders. It was observed that as the workshops progressed, their role gradually changed from demonstrators to companions. In addition to the prescribed function of demonstrating, they accompany and listen to the elders during the workshops and encourage them to share their opinions and suggestions. The staff facilitates dialoguing during the workshops, and after reflecting, the program is improved accordingly.

The present study successfully developed a model for applying IEC workshops to promote CA. After designing the multi-stage program, the team engaged in continuous reflection and review cycles and analyzed the benefits of the program through in-depth interviews with elders, facilitators, and staff. The hope is that more practitioners will join the ranks to serve the elderly in the future and that they will refer to this model in order to achieve CA.

### 6.5. Recommendations 

Several research recommendations are offered based on the collected information as detailed below.

#### 6.5.1. The IEC Workshops’ Resonance Lies in Their Connection to the Local Culture 

The community art project utilized existing community resources integrated its folk customs and landscapes and reflected the local characteristics and emotions through the integration of art and esthetics. The themes of the creative activities centered on an environment and life circle familiar to the participating elders, which induced them to think, identify, and start caring about the things around them. This approach not only made it easy for the elders to feel involved but also created resonance between art and the community, of which the elders form part. 

#### 6.5.2. Attention Must Be Paid to the Reflection Process during Program Implementation and Problems Must Be Remedied Immediately 

Reflection on each workshop session must be emphasized during implementation. The elders should be continuously encouraged to express their opinions and share their suggestions. Where applicable, remedial actions should be taken immediately. If adjustments cannot be immediately applied, the feedback should be used to modify the next round of action. It was noted that most of the elders were concerned about minor details and emotions. Hence, it is critical to pay special attention to the reflections and adjustments at each stage. During implementation, discussions can be held after each round of creation to encourage the elders to fully express their thoughts and feelings. This will facilitate continuous adjustment of the action plan throughout the implementation process, thereby providing the elders with the most appropriate esthetic content based on an up-to-date understanding of their needs.

#### 6.5.3. Need for Empathy Regarding the Elders’ Psyche, Needs, and Feelings during Planning and Implementation 

While the senses diminish with age, mental sensitivity and emotions are heightened. Although the elderly can no longer perceive things as clearly as when they were young, they are able to thoroughly engage in interpersonal relationships. When designing art projects, practitioners often neglect to empathize with elderly people’s psychology, including their feelings, such as their unfamiliarity with new things and their consciousness of their weakened eyesight and hearing. It is therefore necessary to adapt to their mentality and physical state. During the practical components, it is also necessary to consider whether elderly participants’ physical condition will permit them to feel the effectiveness of the completed creations; for instance, an overly complex visual presentation will have a limited effect on those with poor eyesight. Showing empathy and concern and thinking from the elders’ perspective will help them gain a sense of achievement and recognition while experiencing the process of artistic creation.

#### 6.5.4. Sustained Multi-Role Programming Produces Effective Outcomes 

The workshops were conducted with the community’s cooperation for a duration of approximately 6 months. Based on the feedback from the elders’ companions during planning and implementation, the program must be sustained over the long term for the maximum effects to be achieved. A longer program also facilitates the cultivation of closer cooperative relationships with participating elders. For subsequent rounds of implementation, long-term cooperation with the selected community is recommended so that the workshops become part of people’s normal routines. A sustained program also allows practitioners to uncover possibilities within the community.

Implementing the IEC workshops involved interactions and cooperation among community elders, volunteers, care workers, and youth practitioners. However, it was discovered that the community possesses many other resources that could enhance the IEC workshops, such as community experts, farmers, and religious groups. Their involvement would create additional relationships based on mutual learning and integration. Integrating more roles into CA efforts in the future will raise overall community awareness and initiate further interaction among the various actors.

## Figures and Tables

**Figure 1 healthcare-11-00516-f001:**
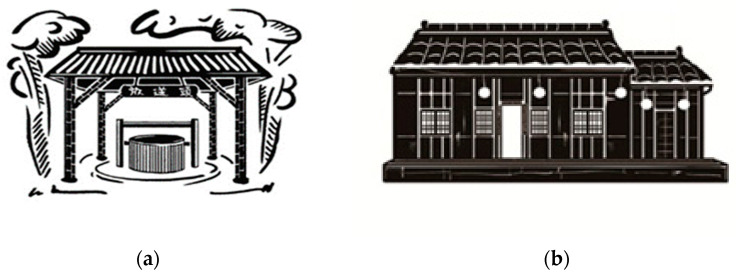
Creations from the Stage 1 IEC workshops. (**a**) Serigraph “Fangsongtou old well” (by E6-C-Week06). In Taiwanese Hokkien, Fangsongtou means “to broadcast using a loudhailer”. The well was named Fangsongtou because it was an important site for information transmission. The elders used to go there to fetch water and do their washing. It was also a playground for children. The community residents recalled the site as a locale for exchanging pleasantries and catching one another up about their current situation and interesting news. (**b**) Serigraph “Great-grandparents’ old house” (by E14-C-Week06). This used to be the residence of a senior who was quite advanced in age. It was built using only tenon joints between bamboo tubes, showcasing high-precision craftsmanship. The internal layout and consideration of daily necessities reflect the life trajectory and historical backdrop at that time.

**Figure 2 healthcare-11-00516-f002:**
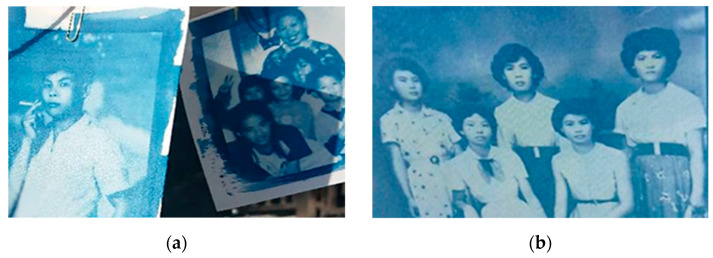
Creations from the Stage 2 IEC workshops. (**a**) Cyanotype under the theme “Tribute to the youthful years” (E2-C-Week12). Photographs of an elder when he was a youth were transformed into cyanotypes, which triggered smiles while recalling the brilliance of life. (**b**) Cyanotype under the theme “Tribute to the youthful years” (E6-C-Week12). A photograph of an elder when she was a youth was transformed into a cyanotype that showcases the luminance and vitality of life.

**Figure 3 healthcare-11-00516-f003:**
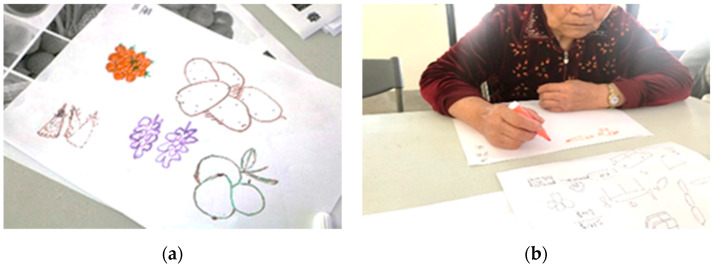
Creations from the Stage 3 IEC workshops. (**a**) Drawing under the theme “Roadside banquet ingredients” (E2-C-Week20). The elders explained the symbolism behind the various food ingredients and recollected happy times on the occasion of the Lunar New Year. (**b**) Drawing under the theme “Roadside banquet ingredients” (E6-C-Week20). The elders gushed eloquently as they recalled the food ingredients.

**Figure 4 healthcare-11-00516-f004:**
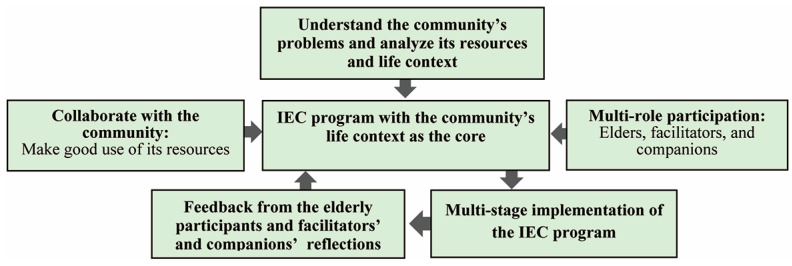
Model for applying IEC workshops to promote sustainable CA.

**Table 1 healthcare-11-00516-t001:** IEC Workshop Framework.

Action Stage	Course Content and Activities	Action Research Implications
**Stage 1 IEC workshops**		
Making serigraphs (silkscreen printing) Three times One time a week 1 August 2021–31 August 2021	Elders and facilitators are introduced and get to know one another. Introduction to serigraphy and provision of instructions on how to do it Distribute materials for serigraphy using cardboard boxes. Assembly of the cardboard boxes. Elders are invited to autograph it. Explain the next class and ask the elders to bring old photos and objects in a cardboard box the following week.	Familiar community landscapes are used as the sources for the designs to give elders greater resonance with their creations.
Opening memory chests for sharing and Reflections Two times One time a week 1 September 2021–21 September 2021	Explain the purpose and content of this workshop. Ask the elders to open their memory chests and take turns discussing the stories behind the old photographs and objects. Intergenerational exchange and sharing for an in-depth understanding of the elders’ life stories Invite the elders to share their feelings about this workshop.	Guide elders to recall their life journey by sharing the stories behind old photographs and objects.
**Stage 2 IEC workshops**		
Creating cyanotypes Three times One time a week 1 October 2021–22 October 2021	Introduction to cyanotypes and provision of instructions on making them Distribute materials and have the elders construct cyanotype postcards according to the steps stated. Complete the production of the cyanotypes.	Help elders recall the various parts of their life through the re-creation of old photographs.
Sharing and reflection prompted by completed cyanotypes Two times One time a week 23 October 2021–6 November 2021	Sharing and appreciation of cyanotypes Invite the elders to share their experiences from the production process and collect feedback for reflection.	Sharing and appreciation of the finished products serve as conversation starters for the elders to initiate communication with their peers and gain a sense of accomplishment.
**Stage 3 IEC workshops**		
Drawing roadside ban-quet ingredients Three times One time a week 7 November 2021–30 November 2021	Explain the content for this stage of the workshops. Discuss traditional festivals and roadside banquet ingredients to create resonance with the elders. As a group, draw the components of roadside banquet dishes from memory.	Guide the elders to recall family reunions on the occasion of special festivals and/or the ingredients grown at home and used for preparing dishes; use food to stimulate resonance; create drawings. Allow the elders to share food-related stories from the past; bring joy to the elders through mutual communication and interactions; relive fond memories of the past.
Make and assemble nostalgic lightboxes displaying symbols Two times One time a week 1 December 2021–14 December 2021	Distribute materials and ask the elders to compose a scene map and, from memory, a collage of their steamboat reunion dinners on the eve of the Lunar New Year. Assemble the lightboxes and place the collages prepared during the previous class into the lightboxes, completing the creations for this stage.	
**Stage 4 IEC workshops**		
IEC esthetics Exhibition One week 15 December 2021–22 December 2021 Interviewing with senior participants, facilitators, and staff 23 December 2021–31 January 2022	Hold an esthetic exhibition to showcase the elders’ artistic creations and life stories. Share information about the workshop implementation process and the elders’ participation. Encourage the elders to interact with friends and relatives in the community. Provide the elders with feedback and bolster their sense of achievement and self-confidence.	The formal exhibition serves to encourage the elders to interact with friends and relatives in the community. Facilitators should also encourage the elders to give feedback by asking them questions.

**Table 2 healthcare-11-00516-t002:** Data Coding.

Research Tool	Example of Code	Meaning of Code
Interviews with the elders	E1-I-Q1-1021	E stands for elder, 1 for the first person, I for interview Q for question number, 1021 for the date (month/day)
Interviews with the facilitators	F1-I-Q1-0107	F stands for facilitator, 1 for the first person, I for interview, Q for question number, 0107 for the date (month/day)
Weekly staff records	S-Week01-01	S stands for staff, Week01 for week number, 01 for document number, EC for elder’s creation
Elders’ creations	EC-Week06	Week06 stands for week number

**Table 3 healthcare-11-00516-t003:** Records of Problems Identified at Each Stage of the Action Research and Corresponding Reflections.

Action Stage	Situation (or Problem) Identified	Response Strategy
Preliminary investigation	The participating elders were relatively quiet. Most participating elders had been involved in agricultural activities.	Sharing/interactive components of the workshops were arranged to encourage the elders to communicate more with one another. Food-related courses were included in the workshops, and these commenced with tasks at which the elders would perform well.
Stage 1	Most of the elders were illiterate. The majority communicate mainly in Taiwanese. Participating elders found executing the steps to be slightly strenuous. The elders were very concerned about color selection. Elders who did not prepare the requested items (old photographs and objects) were disappointed. One of the purposes of having participants bring these items was to prompt them to speak and initiate communication among themselves. The elders were emotionally moved during the sharing sessions, with some experiencing sadness.	Pictures and/or verbal expressions were used when introducing the courses. The steps were designed to be effortless and singular and involved one-time tasks. Thorough investigations were conducted in advance to identify the elders’ likes and preferences. Spare items were kept as extra provisions. Nostalgic elements and extra time for sharing and talking helped kick-start conversations among the elders. Facilitators avoided guiding the elders’ recollection of too many sad stories or images.
Stage 2	The elders could better execute the steps after they were adjusted and simplified. Production speed was fast; the cyanotypes were completed in approximately 1 h and 40 min.	Making cyanotypes was suitable for the elders because the old photographs resonated with them, the process was safe, and the finished products were ready in a short time. For similar workshops in the future, the schedule can be modified to include more time for the elders to interact.
Stage 3	The elders resisted drawing because they lacked self-confidence. The elders spoke enthusiastically about the theme of cooking ingredients. The elders showed a liking for bright colors. The elders enjoyed talking with the youths. The elders were receptive to themes carried over from previous sessions.	Questions were posed to coax the elders’ interest in the art form, and the duplicating method was used to ease them into trying drawing. Given that dialoguing with the elders can generate learning, the facilitators used role reversal to acquire new knowledge from the participants. Bright and vibrant colors should be utilized in future courses and exhibition design in order to reflect the elders’ preferences. Given the community elders’ appreciation of youths’ companionship and care, more time should be spent talking and interacting with them in order to increase emotional exchanges. A slow-paced course model allows elders to participate in a relaxed manner and gain a sense of accomplishment.
Stage 4: Knowledge sharing and holding the exhibition	Most of the elders were shy and slightly awkward during the formal interviews because they lacked confidence, especially in their ability to speak well. The elders’ real needs could be better understood when the interview venues were places very familiar to them. Additionally, the interviewers adopted a chit-chat style to encourage the elders to talk.	When the interviewers used the question cards the research team designed to interview the elders, they paid attention to what the interviewees said they gained or learned from the workshop courses and attempted to guide the elders to provide feedback. The researchers reviewed the elders’ comments and compiled them for use as the basis for improvement during future implementation rounds.

**Table 4 healthcare-11-00516-t004:** On-site IEC workshop records.

**Preliminary Investigations before Workshop Planning**		
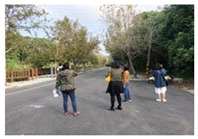	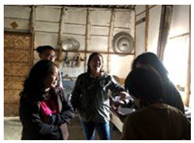	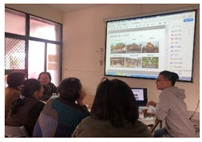
Community road lined with maple trees	Inside an old house in the community	Community meeting and discussion before workshop planning
**Stage 1 IEC workshops: Serigraphy and the opening of memory chests**		
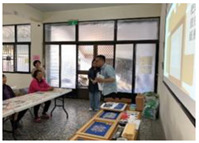	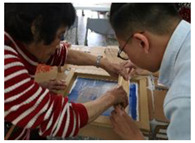	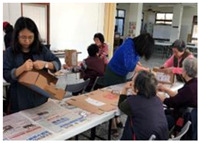
Youths accompanying the elders while providing instructions on serigraphy	Youths and elders attempting serigraphy together	Mounting co-created serigraphs on cardboard boxes
**Stage 2 IEC workshops: Creating cyanotype postcards using old photographs**		
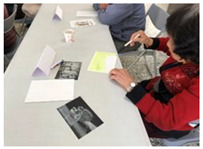	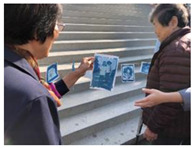	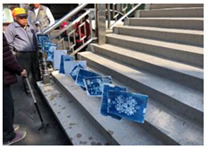
Youths accompanying the elders while creating cyanotypes	Youths accompanying the elders outside to dry the cyanotypes	Youths and elders discussing the co-created cyanotypes
**Stage 3 IEC workshops: Drawing roadside banquet ingredients**		
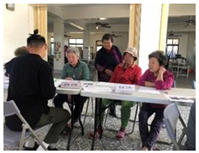	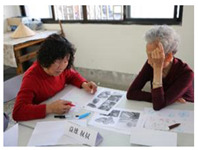	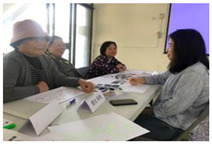
Youths and elders having a discussion before co-creating	Elders conversing before drawing	Youths and elders co-creating photos of food ingredients
**Stage 4 IEC workshops: Holding an esthetic exhibition**		
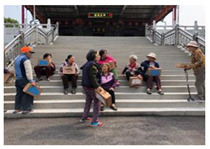	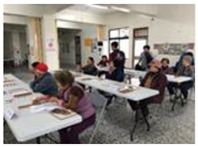	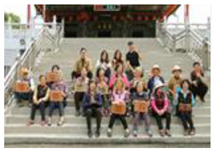
Elders discussing the artwork	Youths and elders interacting during the exhibition	Youth and elderly exhibition participants

## Data Availability

Written informed consent has been obtained from the participants to publish this paper.

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
