# Peer review of "Intergenerational Esthetic Co-Creation Program for Promotion of Community Creative Aging"

_healthcare, 2023, doi:10.3390/healthcare11040516_

Round 1

Reviewer 1 Report

This article makes a good review of the literature on aging from a perspective of health promotion, going beyond what is usually called active aging, promoting the use of art for the development of the person integrated in the community, in a promotion of the person and his social integration . The adoption of the model and its evaluation contribute to the promotion of new intervention perspectives.

Author Response

Thanks for your comments and compliments.

Reviewer 2 Report

The paper of Hsiu Ching Laura Hsieh  and Chun Wei Liu  to be published the major revisions:

The paper is not very original but has been well conducted. I recommend reducing it to a short communication

Author Response

Thanks for your comments. As reviewer 3 requires more details to address the limitations of the paper, the author has to meet the requirements from different reviewers, so it is quite difficult to make it a short communication.

Reviewer 3 Report

This manuscript has a number of strengths.

It is concerned with community creative aging, an important innovation in a world that is increasingly aging. It addresses limitations in this literature. It is global in nature in that it addresses the lack of literature on efforts to promote this phenomenon in Taiwan. While CA is receiving increasing global attention, less attention has been paid to this in the Taiwanese context. It explores the use of esthetics to promote community creative aging. It uses an action research approach, which isa useful contribution. It is effectively grounded in related literature. The manuscript is clearly written and accessible to a wide audience.

With this said, there are some limitations that need to be addressed. Readers are not told when this study took place. It appears to be in 2021 but this is not specified. We are told that the workshops occurred over a period of six months, however the time frame is unclear. For example, were they January-June 2021? Table 1 lists the workshop framework. However, it is unclear how many workshops occurred in each of the four stages. What is the total number of workshops?

Other methodological details are unclear and need to be mentioned. How  elderly participants were interviewed? Under what circumstances were these interviews conducted? Were they at their homes, for example, or some other place in the community? How long were the interviews on average? How many facilitators (volunteers) were interviewed? Under what circumstances? How long were the interviews on average? How many staff (graduate students) were interviewed? Under what circumstances (at community centers? some other location)? How long were the interviews on average?

Finally, what is the nature of the staff records that were reviewed? How did this review process work? What sort of information was gleaned from them? This should be clearer. The use of semi-structured interviews was noted. How were interview schedules developed? We need additional information about this.

Two minor editing issues should be corrected. The line numbers are below:

Line 130: This line should read: "Previous studies have indicated..."
Line 695: Should read "senior participants..."

Author Response

Q1: With this said, there are some limitations that need to be addressed. Readers are not told when this study took place. It appears to be in 2021 but this is not specified. We are told that the workshops occurred over a period of six months, however the time frame is unclear. For example, were they January-June 2021? Table 1 lists the workshop framework. However, it is unclear how many workshops occurred in each of the four stages. What is the total number of workshops?

Author’s Answer:  The above limitations have been addressed, please refer to blue words in table 1 (page6-8). The workshops took place from August 1, 2021 to January 31, 2022 and the venue of the workshop was Hushan community activity center. How many workshops has been held in each of four stage was specified in left column of Table 1. The total number of workshops were 16. Please refer to line 282-283, page 6-8.

Q2: Other methodological details are unclear and need to be mentioned. How elderly participants were interviewed? Under what circumstances were these interviews conducted? Were they at their homes, for example, or some other place in the community? How long were the interviews on average? How many facilitators (volunteers) were interviewed? Under what circumstances? How long were the interviews on average? How many staff (graduate students) were interviewed? Under what circumstances (at community centers? some other location)? How long were the interviews on average?

Author’s Answer:  The semi-structured interview objects for this study contained 15 senior participants, 2 workshop facilitators, and 3 workshop staff. The interview of senior participant was arranged from late December, 2021, to end of January, 2022 (the period after the exhibition at the fourth stage of intergenerational esthetic co-creation workshop). The independent and private lounge in Hushan Community activity center was selected for the interview in December. Each interviewee signed the informed consent before the interview. One person was interviewed at a time, for about 1.5 hours. The interview in January, 2022, was done in 9 senior participants’ home. 9 senior participants expressed that they could more comfortably speak out the feelings at home. The interview questions were based on the questionnaire designed for this study. The questions were designed based on Havighurst’s Activity Theory [12], to understand the psychological, physiological, and social effects of the esthetic workshop on senior participants through the perspectives and ideas after participating in the workshop. (Please refer to blue words between line 301-314).

The interview questions are designed as Appendix A1 ( please refer to blue words between line730-731, p20).

2 facilitators were interviewed in late December, 2021, in the independent and private lounge in Hushan Community activity center. One person at a time was interviewed for about 2 hours. The interview questions were based on the questions designed for this study. The interview questions covered three major dimensions of practical experience, concept identification, and benefit prospect to apply artistic esthetic workshop to implement creative aging. ( please refer to blue words between line 315-320).

The interview questions are designed as Appendix A2. (please refer to blue words between line733-734, p20)

Q3: Finally, what is the nature of the staff records that were reviewed? How did this review process work? What sort of information was gleaned from them? This should be clearer. The use of semi-structured interviews was noted. How were interview schedules developed? We need additional information about this.

Author’s Answer:  3 staff were interviewed as the improvement reference for the planning and execution of next esthetic workshops. The interview was done in the design laboratory (Room DA514) in National Yunlin University of Science & Technology, for about 1.5 hours. The interview content and points contained the appropriateness of content planning to implement creative aging through esthetic workshop, challenge of action practice, and reflection to achieve creative aging. (please refer to blue words between line 321-326).

The interview questions are designed as Appendix A3. (please refer to blue words between line737-738, p21).

Round 2

Reviewer 2 Report

The paper, despite not being very original, was well conducted and deserves to be published.